# Spatially Distributed Light Exposure: Impact on Light Transmission through CAD/CAM Resin-Based Composites of Different Thicknesses

Nicoleta Ilie 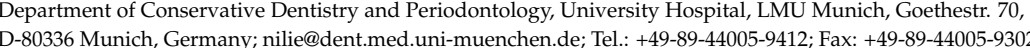

Department of Conservative Dentistry and Periodontology, University Hospital, LMU Munich, Goethestr. 70, D-80336 Munich, Germany; nilie@dent.med.uni-muenchen.de; Tel.: +49-89-44005-9412; Fax: +49-89-44005-9302

**Abstract:** This article reports the variation in incident and transmitted light through four different computer-aided-designed/computer-assisted-manufactured (CAD/CAM) resin-based composites (RBC) of thicknesses up to 4 mm after simulating clinically relevant but non-ideal curing conditions. A violet-blue light curing unit (LCU) was used to simulate 39 different curing conditions for each material and thickness, setting an exposure distance of up to 7 mm in the vertical direction and an additional 13 horizontally varying positions that included a central position and up to 3 mm off-center positions in mesial, distal, buccal, and lingual directions. The data clearly indicate that exposure distance has a stronger influence on the measured light characteristics than the directional and offset deviations from the center position. Increasing exposure distance leveled the differences and should be limited to 3 mm. In all materials, the parameters of the transmitted light follow the pattern of variation of the incident light. The attenuation of light while passing RBCs is high and increases exponentially with thickness to 95–96% of the incident light for 4-millimeter-thick samples. Significant differences in light transmission were observed between the materials, which are well related to chemical composition and refractive index differences between filler and organic matrix. Violet light is still measurable after passing through 4-millimeter-thick RBC layers, but its proportion relative to blue light is drastically reduced.

**Keywords:** light exposure; computer-aided-designed/computer-assisted-manufactured (CAD/CAM) resin-based composites; light transmission; incident light; irradiance

## 1. Introduction

Indirect CAD/CAM (computer-aided-designed/computer-assisted-manufactured) resin-based composite (RBC) restorations are becoming increasingly popular [1]. The main argument for their clinical use is seen in improved polymerization quality through controlled thermal curing under isostatic pressure [2], manifested in improved monomer conversion and density of the polymer network [1,3], and reflected positively in the mechanical [2,4–6] and toxicological behavior [2,7].

CAD/CAM RBC restorations are placed adhesively, with the quality of the bond to the tooth substrate depending on the curing quality of the luting material. Although dual-cure luting materials can be used, they also rely on light exposure, similar to exclusively light-initiated luting materials [8]. This underscores even more the fact that the amount of light that is transmitted through a restoration and is intended to be received by the luting material is crucial for the quality and longevity of the entire restoration.

The importance of the correct handling of light-curing has long been anchored in the consciousness of dentists and is taught and practiced intensively in dental universities [9]. Nonetheless, errors related to incorrect choice and placement of the light curing unit (LCU) during polymerization still occur [10], which are not always attributable to the dentist but often to unfavorable clinical circumstances. The situation became more complex with the use of violet-blue LED (light-emitted diode) LCUs, as it was suspected that the placement

of different LEDs could potentially affect the homogeneity of curing [11]. It is undisputed that the use of light in the violet wavelength range is important not only when using Norish type I photoinitiators such as acrylphosphine oxides with an absorption spectrum in this range [12–14], but also for polymerization initiated by camphorquinone/amine systems, especially under poor curing conditions [15]. The latter aspect was clearly demonstrated both for the effect on the mechanical properties and for the biocompatibility of light-cured RBCs [15]. In this context, it should be noted that under optimal curing conditions, the additional use of violet light does not improve the properties mentioned above compared to blue light polymerization alone [15].

Despite the advantages mentioned, there is a suspicion that the assembly of the LEDs, which usually consists of one violet and two or three blue LEDs, implies an inhomogeneous ratio of violet and blue light that hits the material to be cured. In addition, curing through a material is accompanied by light attenuation that depends on the chemical composition and microstructure of the material [16], the internal porosity [17], anisotropies [18], the thickness, or the restoration geometry [19]. Both monomers [20] and filler particles [17,21], in addition to dyes and pigments [20,22,23], can absorb light. Furthermore, light scattering strongly depends on the size of the fillers [17], with the highest scattering being achieved at sizes approximately half the wavelength of the incident light of the used LCU, that is, approximately 0.2 µm to 0.3 µm [24]. An additional strong factor influencing the scattering is the refractive index matching between fillers and the polymer matrix [25].

In clinical conditions, it is often a challenge to place the LCU perfectly centered and as close as possible to the restoration to be cured. Therefore, in the present study, difficult polymerization conditions or exposure errors are simulated, quantifying the light received from a round surface with a diameter of 3.9 mm that simulates the surface of a restorative material. The LCU is therefore placed up to an exposure distance of 7 mm in the vertical direction and in 1 mm increments for 3 mm decentered in the mesial, distal, lingual, and buccal directions. All these effects are quantified for the incident light and for 4 CAD/CAM RBCs at various thicknesses, up to 4 mm, to simulate the light that a luting material receives when placing such a restoration under ideal and non-ideal curing conditions.

The null hypotheses tested were that irradiance and radiant exposure (total, blue wavelength, and violet wavelength ranges) are independent of (a) exposure distance; (b) deviations in distance and direction relative to the central position; (c) CAD/CAM RBC material; and (d) material thickness.

## 2. Materials and Methods

### 2.1. Specimen Preparation

Four representative CAD/CAM RBCs of the same shade and translucency have been selected (Table 1). Individual specimen sets (n = 6) of plane-parallel test specimens have been prepared for each material in three different thicknesses (0.5 mm, 2 mm, and 4 mm). The width and depth of the samples corresponded to the dimensions of the individual CAD-CAM blocks, which varied between (13.9 and 14.7) mm and (10.6 and 14.6) mm, respectively. CAD/CAM blocks were therefore cut using a low-speed diamond saw (Isomet low-speed saw, Buehler, Lake Bluff, IL, USA) under water cooling. The top and bottom surfaces of the samples were wet-ground in an automatic grinding machine (EXAKT 400CS Micro Grinding System, EXAKT Technologies Inc., Oklahoma City, OK, USA) with silicon carbide abrasive paper (P1200 (600 grit), P2500 (1000 grit), and P4000 (1200 grit)) to achieve the thickness described above with an accuracy of 0.05 mm. A light-emitting diode curing device (LCU, Bluephase® Style, Schaan, Liechtenstein) with a tip diameter of 9 mm was used; the exposure time was 20 s in all 624 analyzed groups.

**Table 1.** Analyzed CAD/CAM RBCs: abbreviation, name, manufacturer, shade, LOT, and composition, as indicated by the manufacturer.

| Code | RBC | Manufacturer | Shade (LOT No.) | Monomer | Filler | |
|------|-----|--------------|-----------------|---------|--------|---|
| | | | | | Composition | wt.% |
| CS | Cerasmart | GC | A3 HT (1702011) | Bis-MEPP, UDMA, DMA | $SiO_2$, barium glass | 71 |
| LC | Luxacam Composite | DMG | A3 (769515) | methacrylates | $SiO_2$-glass | 70 |
| LU | Lava Ultimate | 3M | A3 HT (N933699) | Bis-GMA, UDMA, Bis-EMA, TEGDMA | $SiO_2$, $ZrO_2$, $ZrO_2/SiO_2$ cluster | 80 |
| TC | Tetric CAD | Ivoclar Vivadent | A3 HT (W93631) | Bis-GMA, Bis-EMA, UDMA, TEGDMA | $SiO_2$, barium glass | 71.1 |

Abbreviations: Bis-GMA = bisphenol A glycol dimethacrylate; Bis-MEPP = 2,2-bis(4-methacryloxyethoxyphenyl)propane; Bis-EMA = ethoxylated bisphenol A dimethacrylate; DMA = dimethacrylate; TEGDMA = triethylene glycol dimethacrylate; UDMA = urethane dimethacrylate; $SiO_2$ = silicon oxide (silica); $ZrO_2$ = zirconium oxide (zirconia); wt.% = filler percent by weight.

## 2.2. Spectrophotometry: Incident and Transmitted Light Characteristics

Incident and transmitted irradiances and radiant exposures in the wavelength ranges 300 to 1050 nm have been collected by means of a laboratory-grade, NIST-referenced (National Institute of Standards and Technology) spectrophotometer (USB4000 Spectrometer, MARC (Managing Accurate Resin Curing); Bluelight Analytics Inc., Halifax, NS, Canada). Technical and calibration details of the used system are described in detail elsewhere [26]. The properties measured were related to the size of the detector, which was a circular surface with a diameter of 3.9 mm. The radiant exposure (RE) was specified over the entire wavelength range and additionally differentiated for the blue (430–540 nm) and violet (360–430 nm) wavelength ranges. The measurements were performed with a clinically relevant exposure time of 20 s and were recorded at a rate of 16 measurements/s. A total of 39 different curing positions were simulated for each material and each thickness, corresponding to exposure distances of 0 mm, 3 mm, and 7 mm in the vertical direction and a further 13 horizontally varying positions. The latter includes the central position where the LCU and sensor were placed concentrically and three further distances increasing in 1 mm increments from this central position in all four directions: mesial, distal, buccal, and lingual. To vary the position of the LCU in either a horizontal or vertical position, a mechanical arm was used, which was precisely moved to the intended spatial position. The simulated horizontal positions relative to the spectrophotometer's sensor are summarized in Figure 1.

Light transmission was evaluated by placing CAD/CAM samples of varying thicknesses between the sensor and the curing light while simulating the exposure distances and curing conditions described above. Measurements of incident and transmitted light through three different thicknesses yielded 156 distinct groups for each of the four materials tested.

## 2.3. Statistical Analyses

The normality of the acquired data was confirmed using the Shapiro-Wilk procedure. The effect strength of the parameter's spatial exposure position, CAD/CAM RBC, and specimen thickness, as well as their interaction terms, was assessed by a multivariate analysis (general linear model) with a partial eta-squared statistic. The results were further compared using multiple-way analysis of variance (ANOVA) and the Tukey honestly significant difference (HSD) post hoc test ($\alpha = 0.05$) (SPSS Inc. Version 29.0, Chicago, IL, USA).

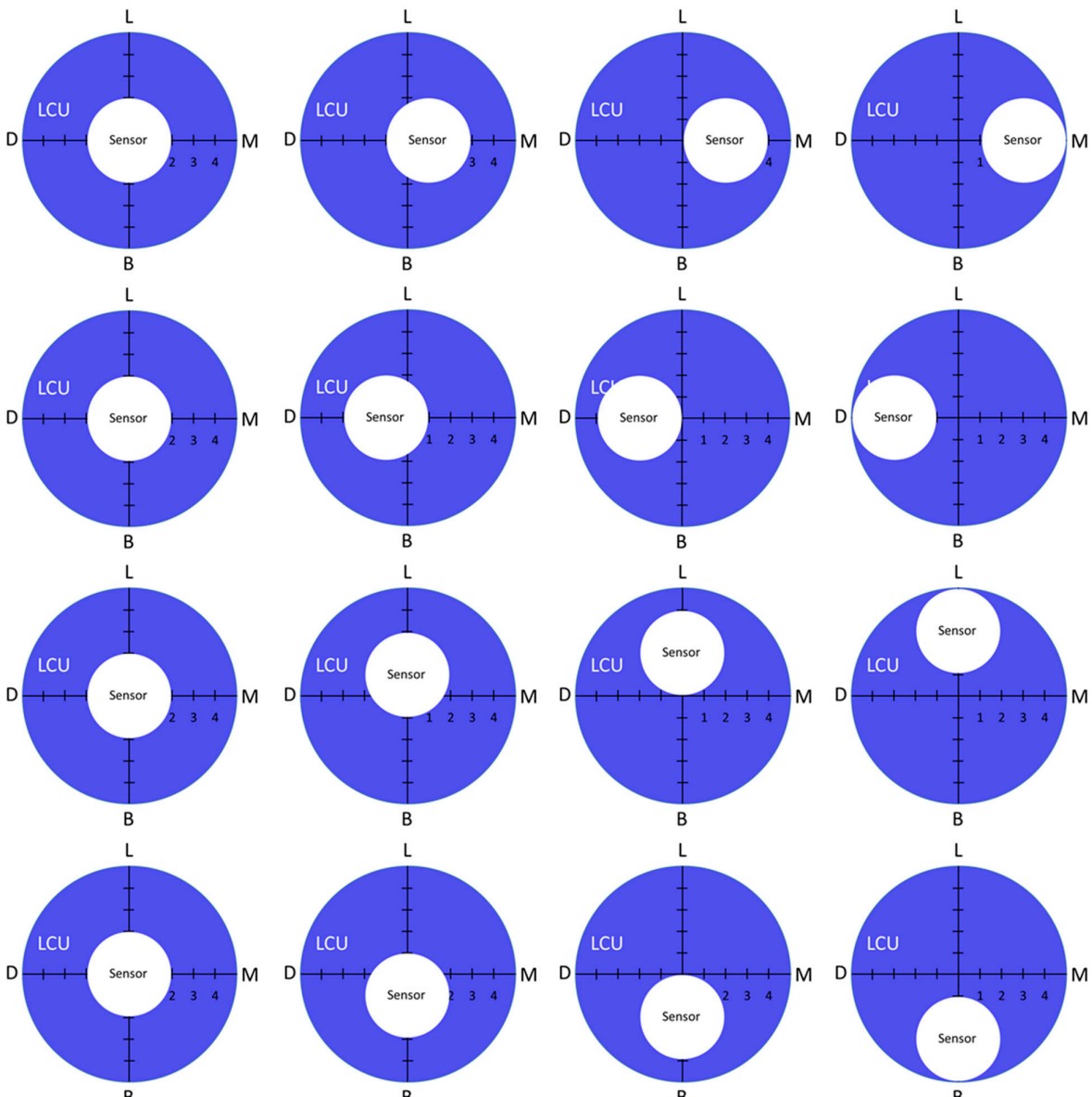

**Figure 1.** The position of the light curing unit (LCU) relative to the spectrophotometer's sensor with a 1 mm incremental displacement up to 3 mm in the mesial (M), distal (D), buccal (B), and lingual (L) directions. The light measured corresponds to the light received by the sensor, i.e., the overlap between the sensor and the LCU in the 13 different horizontal positions. In addition, the LCU was placed at an exposure distance of 0 mm, 3 mm, and 7 mm, which corresponds to a total of 39 spatial variations in light source placement. The diameter of the sensor was 3.9 mm, and the diameter of the LCU was 10 mm.

## 3. Results

### 3.1. LCU Characteristics

The light spectrum of the employed LCU is presented in Figure 2, indicating a violet-blue LCU with peaks at 456 nm (blue wavelength range) and 412 nm (violet wavelength range). The incident light, measured when the LCU is placed directly on the sensor, decreases sharply as it passes through the tested materials, as exemplified by a 2- and 4-millimeter-thick sample, respectively. Detailed data on incident and transmitted irradiance, radiant exposure (RE), and its components in the violet and blue wavelength ranges are summarized in Figures 3–5.

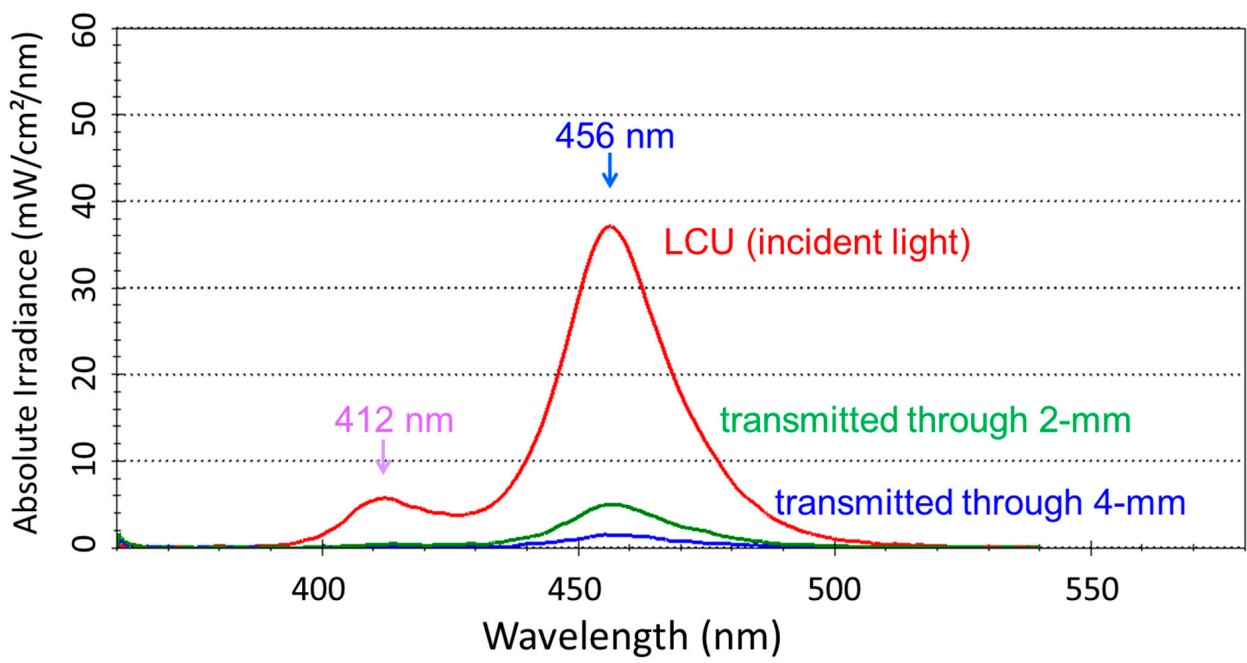

**Figure 2.** An example of the LCU's light spectrum for incident and transmitted irradiance.

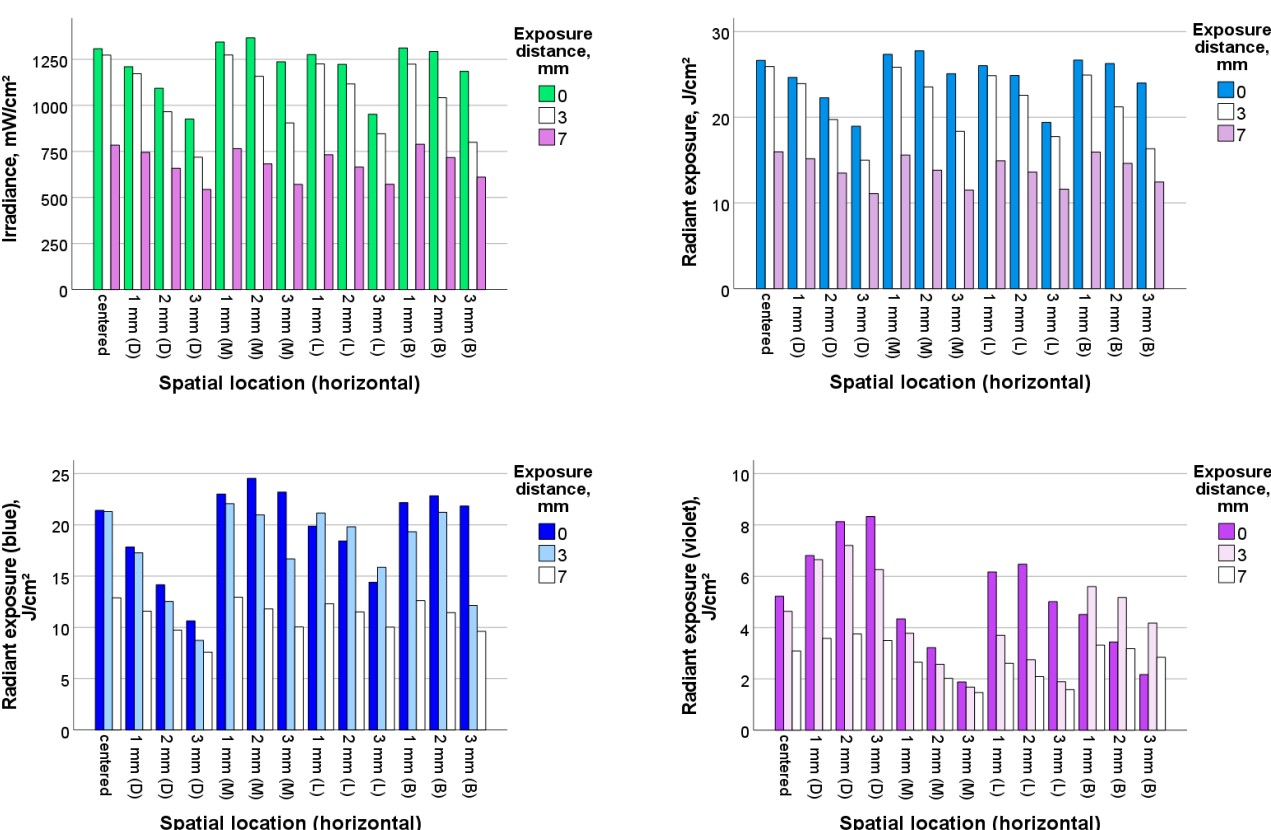

**Figure 3.** LCU: Variation of irradiance and radiation exposure with its subcomponents, violet and blue light, at thirteen different spatial locations and three exposure distances.

### 3.2. Incident Light

The measured light characteristics for the 39 spatially varying positions of the LCU are summarized in Figure 3 in relation to the spectrophotometer sensor. Note that the standard deviations are included but are too small to be visible. The fastest and most progressive

drop in irradiance, total RE, and blue RE with an offset of up to 3 mm from the central position was observed for the distal position (D). The difference is attenuated for the lingual LCU position (L) and small for the mesial (M) and buccal (B) positions. RE in the violet wavelength range behaves in exactly the opposite way, with progressively increasing distal position values. The values measured with direct contact between the LCU and the sensor decrease as the exposure distance increases, but the variation pattern in the horizontal positions remains the same.

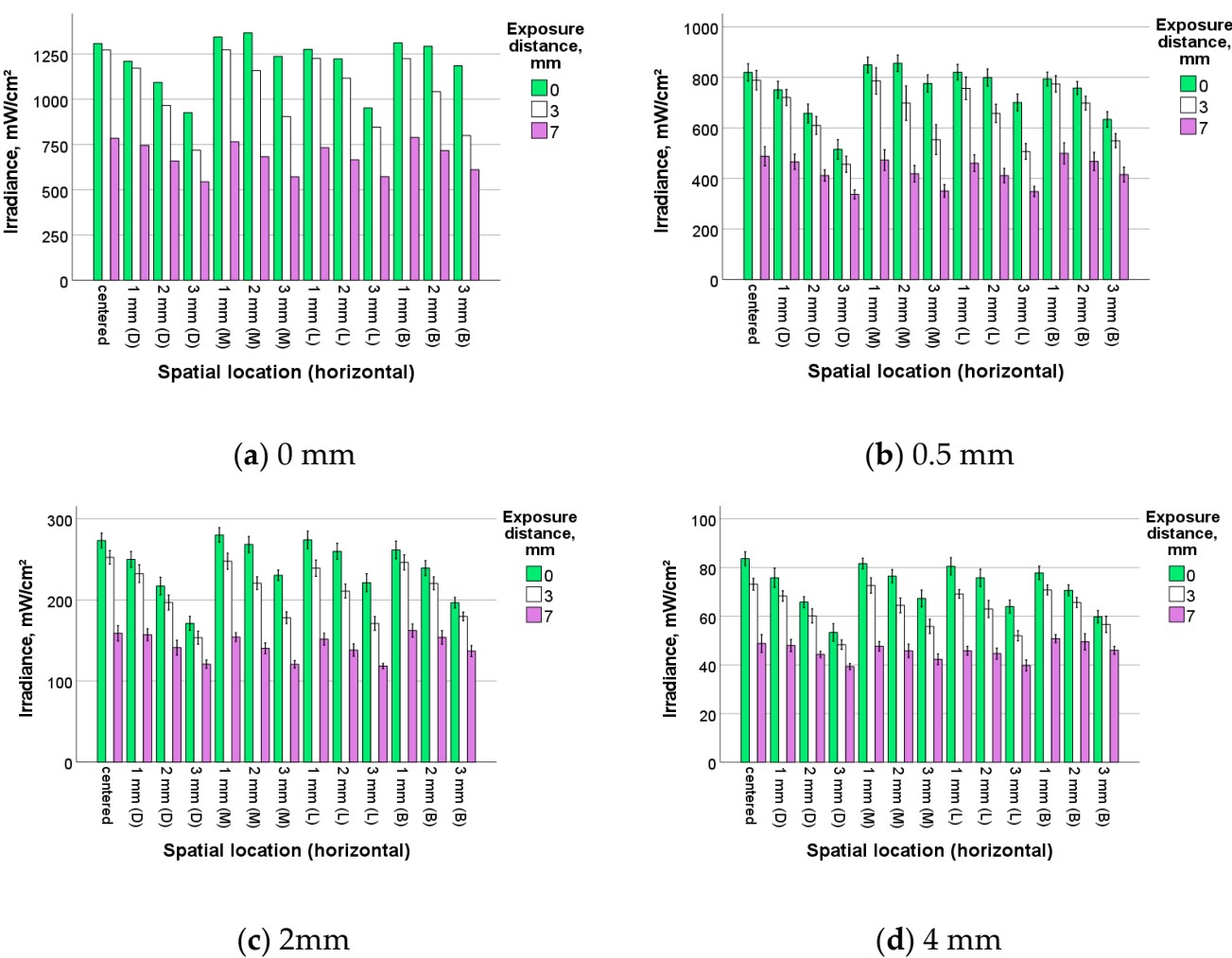

**Figure 4.** Incident (**a**) and transmitted irradiance (**b**–**d**) as a function of spatial location, exposure distance, and material thickness are exemplified for TC.

### 3.3. Transmitted Light through CAD/CAM RBCs

As an example of the four materials measured, Figure 4 summarizes the incident and transmitted irradiances measured on TC samples of 0.5-, 2-, and 4-millimeter thickness. The transmitted irradiance as a function of the 39 different LCU locations follows the pattern of incident irradiance variation shown in Figure 3, with these values being reduced to 65% when passing 0.5-millimeter-thick increments, to 22% in 2-millimeter-thick increments, and to only 6% in 4-millimeter-thick increments.

Direct material comparison in terms of light transmission clearly shows the highest transmitted irradiance, total radiant exposure, and radiant exposure in the blue wavelength range for TC, followed in descending order by CS > LU > LC. In contrast, the highest transmittance in the violet wavelength range was found for CS, followed by LU and TC, which were similar, and finally by LC. The behavior described is summarized in Figure 5 and shown as an example for a material thickness of 2 mm.

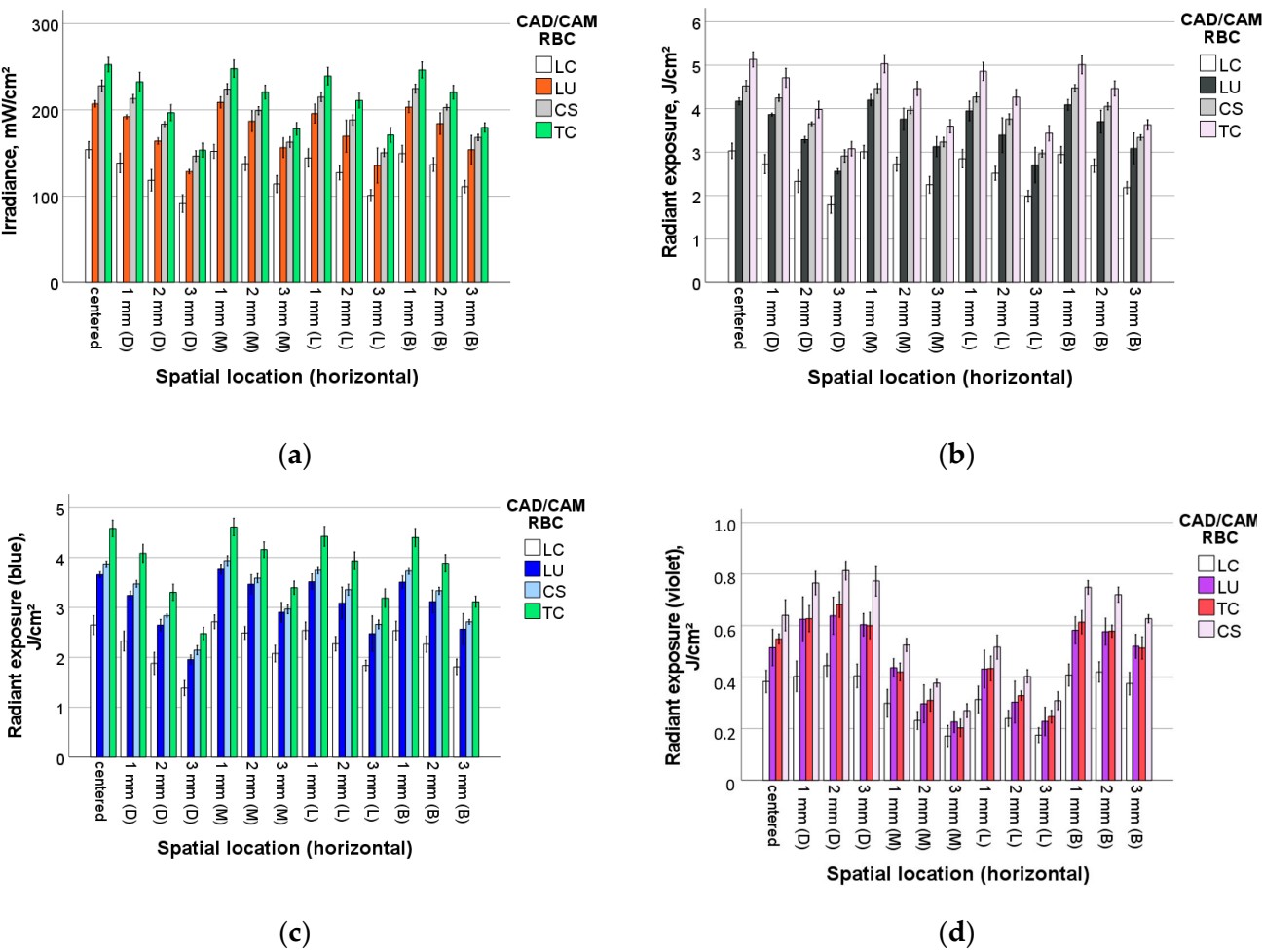

**Figure 5.** Comparison of the four analyzed CAD/CAM RBCs with respect to transmitted light characteristics through a 2-millimeter-thick increment: (**a**) transmitted irradiance; (**b**) transmitted radiant exposure (total); (**c**) transmitted radiant exposure in the blue wavelength range; (**d**) transmitted radiant exposure in the violet wavelength range.

The effect strength of the parameters: direction (mesial, distal, buccal, and lingual), offset (1, 2, and 3 mm from the center in all four directions), and their binary combination on the measured light characteristics is summarized in Table 2. The strength effect of the direction is material- and thickness-dependent. Direction had a very strong effect on the radiant exposure in the violet range ($p < 0.001$; highest partial eta squared values), little on the radiant exposure in the blue wavelength range, and very little on the irradiance and total radiant exposure. Within a measured parameter, the effect strength decreases with thickness. The greatest impact on radiant exposure in the violet region was observed for CS. The greatest impact on irradiance and total radiant exposure was observed for LU. In comparison, the impact of the offset on irradiance, total radiant exposure, and radiant exposure in the blue wavelength range was slightly higher but still small compared to the impact of direction. The impact on radiant exposure in the violet range was small. The combined effect of the direction and offset variables was small and significant only for radiant exposure in the violet wavelength range (Table 2).

**Table 2.** Partial eta squared values of the effect strength of the LCU placement based on the parameter's direction (mesial, distal, buccal, lingual), offset (1 mm, 2 mm, and 3 mm), and their binary combination.

| Material | Thickness | Direction | | | | Offset | | | | Binary |
|---|---|---|---|---|---|---|---|---|---|---|
| | | Irr | $RE_T$ | $RE_V$ | $RE_B$ | Irr | $RE_T$ | $RE_V$ | $RE_B$ | $RE_V$ |
| none | incident | 0.040 | 0.037 | 0.482 | 0.216 | 0.187 | 0.176 | 0.093 | 0.165 | 0.065 |
| LU | | 0.102 | 0.108 | 0.625 | 0.127 | 0.203 | 0.196 | 0.067 | 0.215 | 0.127 |
| CS | | n.s. | n.s. | 0.630 | 0.143 | 0.211 | 0.211 | 0.095 | 0.214 | 0.122 |
| LC | 0.5 mm | 0.045 | 0.045 | 0.533 | 0.181 | 0.161 | 0.161 | 0.104 | 0.157 | 0.073 |
| TC | | 0.051 | 0.048 | 0.550 | 0.207 | 0.175 | 0.173 | 0.104 | 0.173 | 0.079 |
| LU | | 0.061 | 0.061 | 0.534 | 0.157 | 0.238 | 0.239 | 0.136 | 0.242 | 0.087 |
| CS | | n.s. | n.s. | 0.567 | 0.098 | 0.206 | 0.207 | 0.132 | 0.207 | 0.096 |
| LC | 2 mm | 0.049 | 0.051 | 0.525 | 0.135 | 0.240 | 0.240 | 0.129 | 0.239 | 0.086 |
| TC | | n.s. | n.s. | 0.525 | 0.103 | 0.199 | 0.199 | 0.119 | 0.199 | 0.068 |
| LU | | 0.089 | 0.098 | 0.088 | 0.156 | 0.250 | 0.251 | n.s. | 0.268 | n.s. |
| CS | | 0.044 | 0.045 | 0.227 | 0.097 | 0.180 | 0.187 | 0.062 | 0.206 | n.s. |
| LC | 4 mm | 0.060 | 0.065 | n.s. | 0.101 | 0.203 | 0.210 | n.s. | 0.232 | n.s. |
| TC | | n.s. | n.s. | 0.245 | 0.053 | 0.181 | 0.157 | 0.040 | 0.185 | 0.065 |

Abbreviations: Irr = irradiance; $RE_T$ = total radiant exposure; $RE_V$ = radiant exposure in the violet wavelength range; $RE_B$ = radiant exposure in the blue wavelength range; n.s. = the effect is not significant.

## 4. Discussion

Dental light-curing units and the issue of adequate resin-based composite polymerization have a long history of development and are still the subject of ongoing debates [27–29]. If sufficient light does not reach the material to be cured or the emission spectrum of the LCU is not well matched to the absorption spectrum of the photo-initiators, clinically relevant problems must be expected. These deficits are well associated with an increased risk of developing secondary caries [30] in connection with augmented biofilm formation [30] and defective restoration margins [31], as well as reduced adhesion to the hard tooth substance [32]. From a material perspective, improper curing is reflected in a low degree of monomer conversion and insufficient physical and mechanical properties [32–35].

Light curing along with clinical treatment can often result in curing conditions that deviate from the ideal conditions simulated in the laboratory, where the curing device is placed perfectly perpendicular and as close to the surface as possible. Apart from negligent exposure, some negative aspects of light-curing under difficult clinical conditions can be avoided by choosing an appropriate LCU; for example, a pen-like-shaped LCU that mitigates the effect of angulating the incident light in hard-to-reach areas of the posterior region. In contrast, in some clinical situations, it may be difficult to meet the need for a short exposure distance, regardless of the LCU used. An easy-to-understand example of this aspect represents curing a lower first RBC layer in an incrementally placed restoration of a deep cavity, where the cusps impede the LCU's access to the surface to be cured. Aside from the angulation and exposure distance, it can also be difficult to place the LCU perfectly centered on the material, with deviations from the ideal position in the horizontal plane possibly going undetected. This last aspect gains importance with the advent of LED LCUs [36], such as the one analyzed in the present paper, that combine different LED types—blue and violet—with the aim of enlarging the emission spectra of the blue LED LCUs to accommodate, similar to the QTH (quartz tungsten halogen) LCUs, all types of photo-initiators [13,14]. The assumption here would be that the area directly beneath a violet chip does not receive any blue light, and analogously, beneath a blue LED, no violet light, which leads to an inhomogeneous polymerization [11]. All these concerns were simulated in the present study for both incident and through CAD/CAM RBC-transmitted light.

Analysis of the incident light, that is, the light that reaches the sample surface, confirms the expected exponential decrease with increasing exposure distance observed in many LCUs [15]. Up to an exposure distance of 3 mm, the differences in the light properties in all 13 LCU positions in the horizontal plane are very small, indicating a good tolerance of the LCU to such minor deviations from the ideal curing conditions in the vertical direction. A further increase in the exposure distance up to 7 mm leads to an accentuated reduction in measured light characteristics. An important observation is that as the exposure distance increased, the differences between the 13 horizontally varying exposure positions leveled out. To quantify the LCU's performance in the simulated conditions, one needs to define the term adequate polymerization. In light-cured RBCs, it depends on many factors, including light transmission, chemical composition, and the microstructure of the material, which thus controls absorption and scattering, as summarized in the introductory chapter. Given this diversity, it is difficult to give general recommendations on the amount of light required to adequately polymerize a direct restorative material or a luting material through a restoration. Based on a thorough evaluation of many materials, radiant exposure values of 20–24 $J/cm^2$ are recommended to cure 2-millimeter-thick increments of regular RBCs or 4-millimeter-thick increments of bulk-fill RBCs [37,38]. Transferred to the data measured in the present study with an exposure time of 20 s, the tolerance of the LCU to horizontal deviations in the four directions—mesial, distal, lingual, and buccal—up to an offset of 3 mm is consistent only with a 0 mm exposure distance. When the exposure distance increases to 3 mm, all but the 3 mm offset in the distal and lingual directions still meet the defined requirements, while at an exposure distance of 7 mm, all tested conditions are considered insufficient. The clinical consequence of meeting the requirements defined above would be the need to increase the exposure time from 20 to 30–40 s since the radiant exposure was consistently above 10 $J/cm^2$ under all conditions.

In addition to incident light, light attenuation was assessed when passing through indirect restorative materials, with the four analyzed CAD/CAM materials selected to be of the same shade and belonging to the same material category. In fact, differences in the amount of attenuated light were observed, but the profile of the materials that attenuated the incident light related to the analyzed spatial position was similar. This would make it possible to transfer the results of the 39 spatial exposure positions to other materials of this type if only a small number of measurements, possibly only one spatial exposure, are carried out. Moreover, the variation pattern described in the incident light as a function of direction, offset, and exposure distance is directly reflected in the light transmitted through the CAD/CAM RBCs. Since the transmitted light is the remaining light after the incident light has been reflected, absorbed, and scattered, differences in the amount of attenuated light must be related to the microstructure and chemical composition of each material compound and follow an exponential decrease with increasing specimen thickness [39] based on the Beer-Lambert Law. Even if all samples are polished the same and are comparable in type and shade, differences in the amount of reflected light must be considered. For the analyzed materials and similar material categories, the reflected light has been shown to range from 11 to 27% [40].

The most translucent material for blue light in the analyzed materials was TC, followed by CS, LU, and LC. The observed ranking is consistent with the chemical composition of the filler since increased scatter is expected with an increasing refractive index mismatch between the filler and methacrylate matrix [25]. Since the refractive index of the dental monomers that make up the polymer matrix of the RBCs is about 1.55, it represents a better match for the fillers used in TC and CS, which are essentially silica (refractive index n = 1.4527) and barium glass (n = 1.5100), compared to LU, which contains, in addition to silica, elements of higher atomic numbers such as zirconium (n = 2.1326). Small differences in translucency between TC and CS at comparable filler loading and chemistry can be related to the slightly larger fillers observed in TC [41]. A larger filler with a similar amount of filler results in a lower filler/matrix interface, with the consequence of lower scattering and higher light transmission [42]. For LC, it was shown that the filler system contains

neither Ba nor Zr but Al and Si [41] and is apparently based on an aluminosilicate mineral and not on an aluminosilicate glass. The higher light attenuation is an indication of a higher refractive index compared to the aluminosilicate glass in TC and CS.

Light attenuation while passing through a medium is wavelength-dependent, with shorter wavelengths being attenuated to a greater extent than longer wavelengths. Although violet light is still measurable after passing through 4-millimeter-thick RBC layers, its proportion relative to blue light is drastically reduced. To put this statement into numbers, for the transmitted light in LU, the ratio of violet light to blue light at ideal exposure was 20% for the incident light, decreasing to ratios of 17% (0.5 mm), 12% (2 mm), and 11% (4 mm) when passing through specimens of different thicknesses.

Finally, it can be stated that all null hypotheses are rejected. Given the clinical implications of the results of this study, it is strongly recommended that the LCU tip be centered over the restoration and that the exposure distance be less than 3 mm. Curing situations that deviate from these values must be compensated for by longer exposure times.

## 5. Conclusions

A significant dependence of the position of the LCU on the amount of light received by a restoration was found, with the influence of the exposure distance being stronger than the influence of directional and offset deviations from the central position in the horizontal plane. In addition, differences in the variations in the horizontal plane were balanced with increasing exposure distance. The attenuation of light while passing RBCs is high and increases exponentially with thickness to 95–96% of the incident light for 4-millimeter-thick samples. The most translucent material for the blue wavelength range was TC, while CS allows better transmission of violet light.

**Funding:** This research received no external funding.

**Data Availability Statement:** Data is available on request.

**Conflicts of Interest:** The author declares no conflict of interest.

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
