# Peer review of "Spatially Distributed Light Exposure: Impact on Light Transmission through CAD/CAM Resin-Based Composites of Different Thicknesses"

_jcs, doi:10.3390/jcs7090391_

Round 1
Reviewer 1 Report
1. The "Introduction" part, unfortunately, is completely missing. An exhaustive literary review of the topic under consideration should be given in the article.
2. All abbreviations must be disclosed, in particular "CAD/CAM".
3. In paragraph 2.1, you indicate only the thickness of the samples, but what was their planar size?
4. It is not entirely clear what meaning the change in the horizontal position of the light beam has. Wouldn't it be better to study curing in a situation where the light beam is directed at an angle in the surface; can this simulate a clinical situation where it is difficult to achieve a completely normal incidence of a beam of light on a surface to be cured?
5. Is it possible to estimate the extinction coefficient of blue light when passing through various cured composites?
6. "Analysis of the incident light, that means the light that reaches the sample surface, confirms the expected exponentially decrease [37] with increasing exposure distance, based on the Beer-Lambert Law." (p. 9)
This is a dubious statement. As you know, according to the Lambert-Beer law, the attenuation of light occurs due to its absorption. However, the absorption of blue light in air is very low and is unlikely to be significantly attenuated at distances of a few mm. In your case, the weakening of the light intensity (I) with increasing distance to the sample may be due to beam divergence. For example, for a point light source, such a dependence has the form I ~ 1/R2, where R is a exposure distance. In any case, you should study the dependence of irradiation on the exposure distance, taking more than 4 distances for this, and analyze this experimental dependence, whether it will be exponential or some other.
No
Author Response
All comments to the corresponding author have been addressed independently below. The author’s rebuttal is always in BLUE and where changes have been added to the revised manuscript in light of the reviewer's comments these are presented in RED.
The author would firstly like to thank the reviewers for taking the time to read and critically appraise the manuscript and secondly to thank the reviewers for their positive constructive comments in improving the work.
Reviewer1:
Comments and Suggestions for Authors
1. The "Introduction" part, unfortunately, is completely missing. An exhaustive literary review of the topic under consideration should be given in the article.
Author’s response: I have to strongly disagree with the reviewer on this comment. The introduction summarizes exactly what the study aimed to do. The study quantifies the effects of incorrect polymerization in terms of light transmission, i.e. it simulates an extremely high number of polymerization conditions. It determines the incident and the transmitted light, exemplified on 4 CAD/CAM RBCs of different thicknesses. In addition, it determines all these aspects for a violet-blue LED LCU. It is precisely these aspects that are discussed in the introduction: clinically incorrect exposure, which arises either from ignorance or from clinically given situations; the issue with violet-blue LEDs, the meaning of violet light in the polymerization of methacrylate dental materials, the consequences of incorrect exposure, the factor that determines light attenuation in a material. The second part of the comment, which I also strongly disagree with, is the requirement for a comprehensive literature review in the introductory section - this is not part of a research article, but a review. The introduction reflects the current status of the topic and describes the need for the following study. I hope that the reviewer agrees with these aspects.
2. All abbreviations must be disclosed, in particular, "CAD/CAM".
Author’s response: I would like to thank you for this comment and have spelled out the abbreviation accordingly in the revised manuscript. The abbreviation CAD/CAM was now explained when it was first used - in the abstract - and not only in the introduction, as was the case in the original manuscript. Once explained, only the abbreviation is used. The abbreviation QTH was not explained in the discussion and was added in the revision. I apologize for this omission.
3. In paragraph 2.1, you indicate only the thickness of the samples, but what was their planar size?
Author’s response: Thanks for this comment. The dimensions of the samples corresponded to the size of the CAD/CAM blocks. This exact information has been noted in the revised manuscript.
4. It is not entirely clear what meaning the change in the horizontal position of the light beam has. Wouldn't it be better to study curing in a situation where the light beam is directed at an angle in the surface; can this simulate a clinical situation where it is difficult to achieve a completely normal incidence of a beam of light on a surface to be cured?
Author’s response: What was simulated here is the situation when the LCU is not placed centered on the restoration to be cured but with an offset. This is exactly the aim of the study: since the violet and blue LEDs are placed in a given geometry (2 blue and one violet) it is expected that the off-set from the central position will result in a higher or lower proportion of blue/violet light, corresponding to the placement of the chips. The assumption was that the area directly beneath a violet chip does not receive any blue light and analogously beneath a blue LED no violet light, which leads to an in-homogeneous light depending on the relation between the specimen, represented here by the sensor, and curing unit.
Angulation would be an additional parameter to test. Please note, however, that 39 positions for 4 situations (incident light/transmitted light) in each of the 4 materials examined resulted in 156 test groups. This study design is more than comprehensive.
5. Is it possible to estimate the extinction coefficient of blue light when passing through various cured composites?
Author’s response: This was not the aim of the present work. Yes, it is possible to measure the absorption coefficient of the materials as a function of the wavelength as we did in the cited papers. For accurate calculations, however, more than 3 thicknesses are required. Our goal here was to evaluate the effects of non-ideal, but clinically relevant placement of the LCU.
6. "Analysis of the incident light, that means the light that reaches the sample surface, confirms the expected exponentially decrease [37] with increasing exposure distance, based on the Beer-Lambert Law." (p. 9)
This is a dubious statement. As you know, according to the Lambert-Beer law, the attenuation of light occurs due to its absorption. However, the absorption of blue light in air is very low and is unlikely to be significantly attenuated at distances of a few mm. In your case, the weakening of the light intensity (I) with increasing distance to the sample may be due to beam divergence. For example, for a point light source, such a dependence has the form I ~ 1/R2, where R is a exposure distance. In any case, you should study the dependence of irradiation on the exposure distance, taking more than 4 distances for this, and analyze this experimental dependence, whether it will be exponential or some other.
Author’s response: Absolutely correct. Lambert-Beer's law was intended to be mentioned for the transmitted light, not for the incident light, as the cited paper also indicates. I apologize for the misplacement of the statement in the discussion of incident light and thank you for the vigilance. Furthermore, simulating more curing conditions would certainly refine the data – however, the present study was designed to simulate clinically relevant conditions and to indicate the trends. A study with 624 groups and almost 4000 measurements, as the present one, is certainly not a simple study design.
Reviewer 2 Report
This study is of practical interest because it can help the dentist choose the material for the restoration of teeth. However, the manuscript needs significant revision.
The article lacks an Introduction. At the same time, references to articles are provided.
1. A description of the current state of research in this field of dentistry is required. Indicate what is the relevance and scientific novelty of the study compared to previously published works.
Materials and methods
2. Specify the characteristics of the used violet-blue LED, the grit of the sanding paper.
3. Did the authors additionally use multiple-way ANOVA test with partial eta-squared statistics (ηp²)? If not, please explain why.
Results and discussion
4. It is necessary to study the parameters of light attenuation when passing through the used restorative materials, in particular transmittance T, absorbance A, opacity O.
5. There is no explanation for the results shown in Figures 3, 4, 5 for various dental photocurable restorative materials CS, LC, LU, TC. What is the reason for the significant decrease in radiant exposure for the distal position compared to other positions? (page 5, figure 3).
6. On page 10, the word aluminosilicate is repeated twice.
Conclusions
7. Which of the considered dental restorative materials is the most promising for violet-blue light curing and why? Please specify.
Minor editing of English language required
Author Response
All comments to the corresponding author have been addressed independently below. The author’s rebuttal is always in BLUE and where changes have been added to the revised manuscript in light of the reviewer's comments these are presented in RED.
The author would firstly like to thank the reviewers for taking the time to read and critically appraise the manuscript and secondly to thank the reviewers for their positive constructive comments in improving the work.
Reviewer2:
Comments and Suggestions for Authors
This study is of practical interest because it can help the dentist choose the material for the restoration of teeth. However, the manuscript needs significant revision.
Author’s response: Thank you for the appreciation.
The article lacks an Introduction. At the same time, references to articles are provided.
1. A description of the current state of research in this field of dentistry is required. Indicate what is the relevance and scientific novelty of the study compared to previously published works.
Author’s response: The introduction summarizes exactly what the study aimed to do. The study quantifies the effects of incorrect polymerization in terms of light transmission; it simulates an extremely high number of polymerization conditions. It determines the incident and the transmitted light, exemplified on 4 CAD/CAM RBCs of different thicknesses, for a violet-blue LED LCU. It is precisely these aspects that are discussed in the introduction: clinically incorrect exposure, which arises either from ignorance or from clinically given situations; the issue with violet-blue LEDs, the meaning of violet light in the polymerization of methacrylate-based dental materials, the consequences of incorrect exposure and he factors responsible in a material for light attenuation. In fact, the current state of research was clearly described in the introduction.
Materials and methods
2. Specify the characteristics of the used violet-blue LED, the grit of the sanding paper.
Author’s response: LCU characteristics are the results of this study and are presented in detail in paragraph 3.1. and 3.2 – Please consider the manuscript text.
The roughness of the silicon carbide abrasive paper is presented in chapter 2.2 “Top and bottom surfaces of the samples were wet-ground in an automatic grinding machine (EXAKT 400CS Micro Grinding System EXAKT Technologies Inc. OK, USA) with silicon carbide abrasive paper (P1200, P2500, P4000) to achieve the thickness described above with an accuracy of 0.05 mm.” I assume that the reviewer would like to know the equivalence between FEPA and ANSI, i.e. between P and Grit. I have included the P numbers because this is the manufacturer's only indicator of the sandpaper used, as the equivalence between FEPA and ANSI is not always stated. But I understand if some readers are more familiar with FEPA, and I also indicated the grit in parentheses now.
3. Did the authors additionally use multiple-way ANOVA test with partial eta-squared statistics (ηp²)? If not, please explain why.
Author’s response: Yes, a multivariate analysis (general linear model) with partial eta-squared statistics was used, as described in chapter 2.3.
Results and discussion
4. It is necessary to study the parameters of light attenuation when passing through the used restorative materials, in particular transmittance T, absorbance A, opacity O.
Author’s response: Transmittance, absorbance, and opacity are all parameters that are calculated from the incident and transmitted irradiance, the parameters measured in the present study, by simple formulas (T = ratio of transmitted irradiance to incident irradiance; Absorbance (A) = − log(T), etc), but these optical parameters are not relevant for the present study. This will add 468 values for each optical parameter - T, A, and O - to the paper. The paper actually aims to quantify the transmitted irradiance within the simulated curing conditions in view of the variation in light that may potentially reach an underlying luting material.
5. There is no explanation for the results shown in Figures 3, 4, 5 for various dental photocurable restorative materials CS, LC, LU, TC. What is the reason for the significant decrease in radiant exposure for the distal position compared to other positions? (page 5, figure 3).
Author’s response: Please consider the exhaustive explanation made in the discussion, based on the microstructure and composition of the materials. The reason for the differences in medial-distal-lingual-buccal position is the LED placement in the analyzed LCU, which consists of 2 blue and one violet LED. Their location affects the result, as the light is not emitted homogeneously within the entire light guide. The aim of the study was in fact to evidence and quantify this effect. The hypothesis was that the area directly beneath a violet chip does not receive blue light and analogously beneath a blue LED no violet light – so the total amount of blue and violet light reaching the simulated filling (the sensor) would vary.
6. On page 10, the word aluminosilicate is repeated twice.
Author’s response: What was meant here is once the aluminosilicate (thus a mineral) and once the aluminosilicate glass (amorphous state) and is in fact not a repetition.
Conclusions
7. Which of the considered dental restorative materials is the most promising for violet-blue light curing and why? Please specify.
Author’s response: Thank you for this comment - The revised conclusions mention now also the materials with the best light transmission for blue and violet light.
Round 2
Reviewer 1 Report
Thank you for adding the Introduction part in the revised version.
The article can be published as present.
Author Response
Author’s response: Thank you for the appreciation. Unfortunately, I wasn't aware that a part was missing from the submitted manuscript. It must have been a software error during the format change.
Reviewer 2 Report
Dear author, please note that the original version of the manuscript was completely missing the text of the introduction.
At the moment, the manuscript has been significantly improved and requires minor modifications.
Section 2.3 should have mentioned multivariate analysis (general linear model) with partial eta-squared statistics.
In the Materials and Methods section, indicate the manufacturer and country of the LCU, as well as the tip diameter and light source type.
The links are formatted carelessly, in particular, there are many typos when indicating DOI.
Minor editing of English language required
Author Response
Reviewer2, revision 2:
Comments and Suggestions for Authors
Dear author, please note that the original version of the manuscript was completely missing the text of the introduction.
Author’s response: Thanks for the clarification. Unfortunately, I wasn't aware that a part was missing.
At the moment, the manuscript has been significantly improved and requires minor modifications.
Author’s response: Thank you for the appreciation.
Section 2.3 should have mentioned multivariate analysis (general linear model) with partial eta-squared statistics.
Author’s response: Thank you for the recommendation – the term was added to the revision.
In the Materials and Methods section, indicate the manufacturer and country of the LCU, as well as the tip diameter and light source type.
Author’s response: Thank you for reporting this omission. The information is now added to the revision.
The links are formatted carelessly, in particular, there are many typos when indicating DOI.
Author’s response: The references have been checked – thank you for pointing out the omission. Please consider that thank not all references have a DOI – in a few of them the PMID is indicated for identification. This was the reason why the software did not recognize them and omitted to indicate the DOI.
Round 3
Reviewer 2 Report
The manuscript may be published.